# Modeling Study of the Effects of *Ageratum conyzoides* on the Transmission and Control of Citrus Huanglongbing

**DOI:** 10.3390/plants12203659

**Published:** 2023-10-23

**Authors:** Ying Wang, Shujing Gao, Yujiang Liu, Huaiping Zhu

**Affiliations:** 1Key Laboratory of Jiangxi Province for Numerical Simulation and Emulation Techniques, Gannan Normal University, Ganzhou 341000, China; wy100126592@163.com (Y.W.);; 2LAMPS and CDM, Department of Mathematics and Statistics, York University, Toronto, ON M3J 1P3, Canada; huaiping@yorku.ca

**Keywords:** Huanglongbing, *Ageratum conyzoides*, mathematical model, basic reproduction number, transmission, paradoxical effect

## Abstract

*Ageratum conyzoides* (*A. conyzoides*) is commonly found or intentionally planted in citrus orchards due to its ability to provide habitat and breeding grounds for the natural enemies of citrus pests. This study aims to expand from a switching Huanglongbing model by incorporating the effects of *A. conyzoides*, vector preferences for settling, and pesticide application intervals on disease transmission. Additionally, we establish the basic reproduction number $R_{0}$ and its calculation for a general switching compartmental epidemic model. Theoretical findings demonstrate that the basic reproduction number serves as a threshold parameter to characterize the dynamics of the models: if $R_{0}<1$, the disease will disappear, whereas if $R_{0}>1$, it will spread. Numerical results indicate that the recruitment rate of *A. conyzoides* not only affects the spread speed of Huanglongbing but also leads to paradoxical effects. Specifically, in cases of high infection rates, a low recruitment rate of *A. conyzoides* can result in a decrease, rather than an increase, in the basic reproduction number. Conversely, a high recruitment rate can accelerate the spread of Huanglongbing. Furthermore, we show how different vector bias and pesticide spraying periods affect the basic reproduction number.

## 1. Introduction

Huanglongbing (HLB) or citrus green disease is the most prevalent, dangerous, and devastating disease for citrus almost worldwide. The Asian citrus psyllid (ACP, *Diaphorina citri Kuwayama*) is a principal vector transmitting the bacterium, commonly known as *Candidatus Liberibacter asiaticus* (*Las*) [1,2] in a persistent, circulative, and propagative manner. Because there is no known available cure for HLB [3], disease prevention is more crucial than treatment controlling in HLB-endemic regions [4]. Currently, prevention of HLB has focused primarily on effective control of the ACP to further reduce the spread of pathogens [5].

The citrus psyllid belongs to the family *Psyllidae* of the order *Hemiptera* and is an important pest during the new shoot period of plants in the *Rutaceae* family, mainly including *Citrus reticulata Blanco*, *Citrus maxima Merr.*, *Citrus sinensis Osbeck*, and *Murraya paniculata Jack* [6,7]. Historically, biologists have limited the host range of the citrus psyllid to plants in the *Rutaceae* family. The current monitoring and control of ACPs and HLB disease is also limited to *Rutaceae* plants. However, recent research has shown that ACPs can inhabit and feed on non-host plants outside of the *Rutaceae* family [8]. These *non-Rutaceae* plants can serve as temporary refuges for ACPs, and create conditions for their long-distance migration and spread, which subsequently affects the accurate monitoring and effective control of ACPs and HLB disease [9]. Field investigations have found that, without human interference, adult ACPs could stay for long periods of time on common weeds in citrus orchards, such as *Ageratum conyzoides*, *Eupatorium catarium*, and *Datura* [10]. It was reported in [9] that the longest survival time of adult ACPs on *A. conyzoides* was 48 days at an average temperature of $35{\;}^{\circ }C$ under adverse conditions such as pesticide application or citrus death.

*A. conyzoides*, a weed belonging to the *Asteraceae* family, is commonly found in citrus orchards and their surrounding areas. A certain amount of *A. conyzoides* is intentionally retained or planted in citrus orchards to provide habitat and breeding sites for citrus mites, the natural enemies of pests. However, *A. conyzoides* can interfere with the efficacy of pesticide control of ACPs and significantly impact the development of psyllid populations [11]. Firstly, *A. conyzoides* serves as a host plant of ACPs, providing them with necessary food and habitats for reproduction and survival. The abundance of *A. conyzoides* in citrus orchards attracts and nurtures ACPs, leading to increased density and distribution range. Secondly, *A. conyzoides* creates a protective environment that shields ACPs from pesticide spraying. ACPs seek refuge within weeds, making it more challenging for them to be affected by pest control practices. Experimental results [9] demonstrate that, after spray treatment of 1.8% Avermectin EC 90 mg/L, the adult ACP mortality rate reached 79.49% without *A. conyzoides*, whereas in the presence of *A. conyzoides*, the adult mortality rate dropped to 42.76%, which shows the clear impact of *A. conyzoides* on ACP control.

Currently, pesticide spraying is an essential component of ACP control and has been considered one of the most effective methods for controlling ACPs. Different pesticides have different persistence periods, which depend on factors such as their chemical composition, application method, and environmental conditions. Pesticides used to control citrus psyllids typically have durations of effectiveness ranging from a few days to a few weeks. To ensure continuous control of ACPs, the pesticides often require frequent applications. The pesticide application interval consists of two phases: the effectiveness period and the non-effectiveness period. Field experiment results [9] indicated that the selection ratio of adult ACP was 26.73% for *A. conyzoides* during the effectiveness period, and then there was only 1.24% during the non-effectiveness period. Adult ACPs exhibit different settling preferences on citrus trees and *A. conyzoides* during the effectiveness and non-effectiveness periods of pesticides.

Mathematical models have played an important role in understanding the epidemiology of vector-transmitted plant pathogens, in particular viral pathogens [12,13,14,15]. Mathematical models of HLB disease has mainly focused on comprehensive control measures [16,17], incubation or latent period [18,19], pesticide resistance of ACPs [20], and climatic factors [12,21,22]. However, the effect of the new host selection mechanism on the population of ACPs and spread of HLB is absent. To assess the risk of the spread of the host spectrum of “citrus psyllid - HLB”, we propose a general switching dynamic model to investigate the interference of *A. conyzoides* on the ACP population and HLB transmission. We then define the basic reproduction number $R_{0}$ and present the analytical results for calculating the number from the general switching compartmental epidemic model. We then derive the implicit expression for the basic reproduction number of the HLB switching model. The threshold dynamics will be discussed in terms of the basic reproduction number to evaluate the impact of several key factors, including the recruitment of *A. conyzoides*, vector preferences for hosts and weeds, and pesticide application intervals, on the development of the ACP population and HLB transmission. The dynamical analysis and simulations of our models will yield some new insights into the comprehensive control of ACPs and the effective containment of HLB, and also provide some useful guidance to orchard managers on the quantity of *A. conyzoides* essential to retain and level of pesticide spraying.

## 2. Methods

### 2.1. Model Formulation

The persistence period of pesticides refers to the duration required for pesticides to exhibit their insecticide fungicidal or herbicidal effects in crops or soil, It is important to note that each pesticide has a special persistence period. During the persistence period of the pesticide, some adult ACPs in an orchard would leave citrus trees and land and settle on *A. conyzoides*. However, after the pesticide becomes ineffective, the vast majority of ACPs would quickly return to their host plants. In this section, we aim to establish a switching Huanglongbing epidemic model that describes the interaction among host plants (citrus trees), non-host plants (*A. conyzoides* weeds), and vectors (ACPs). This model takes into account the changes in settling preferences of ACPs on both host and non-host plants under different periods.

Our study focuses on a whole citrus orchard. We denote $N_{h}$ as the total number of citrus trees, which is further divided into susceptible (healthy) trees $S_{h}$ and infected trees $I_{h}$. Let $N_{v}$ be the total number of ACPs, which is divided into susceptible and infected ACPs in the citrus trees $X_{c}$ and $Y_{c}$, and susceptible and infected ACPs in the *A. conyzoides*$X_{w}$ and $Y_{w}$, respectively. Let *W* be the number of *A. conyzoides* in the orchard. The model in Figure 1 describes the dynamic of ACPs *A. conyzoides* and trees with three different types of reservoirs. In order to explore how the pathogen is transmitted between trees and ACPs, we provide some details and assumptions of the model with equations.

We assume that all newly planted citrus trees are susceptible, and the immediate replanted measure is implemented in the orchard; therefore, $N_{h}$ remains a constant denoted by *K*. We know that ACPs only lay eggs and reproduce on citrus trees, and $\Lambda_{v}$ is the constant recruitment rate for ACPs.

The healthy trees would be inoculated by the viruliferous ACPs on the trees, and the non-viruliferous ACPs on the trees would acquire the virus from infected trees. The forms we adopt for the overall rate at which uninfected trees become infected and the overall rate at which non-viruliferous ACPs on the trees become viruliferous would be
$$\frac{\beta_{1}S_{h}Y_{c}}{N_{h}}\;\;\mathrm{and}\;\;\frac{\beta_{2}X_{c}I_{h}}{N_{h}},$$
respectively, where $\beta_{1}$ is the probability that a susceptible citrus tree becomes infected from contact with viruliferous ACPs, $\beta_{2}$ is the probability that a non-viruliferous ACPs becomes viruliferous from contact with an infected citrus tree.

We consider the case that the total number of citrus trees remains a constant, and the diffusion rate of ACP from *A. conyzoides* to trees is assumed to be constant δ. However, as we know, the dispersal rate of ACPs from citrus trees to *A. conyzoides* would increase with the increase in the number of weeds. We adopt the saturated dispersal forms [23,24]
$$\frac{\delta \alpha W}{1+\alpha_{1}W},$$
where α represents the contribution of individuals to population growth, while $\alpha_{1}$ represents the inhibitory effect of resource scarcity on population growth. δ is the diffusion rate of ACPs.

Currently, spraying pesticides to kill ACPs is still the most effective method to control HLB disease. Each pesticide has a certain persistence period. Assuming the effectiveness period is $T_{1}$, and the non-effectiveness period is $T_{2}$, then $T=T_{1}+T_{2}$ is the pesticide application interval. Simply, we assume the pesticide is applied at time point kT ($k\in Z^{+}$, $Z^{+}$ denotes a non-negative integer set); therefore, $\left(kT,\;kT+T_{1}\right]$ is the duration of effectiveness, and $\left(kT+T_{1},\;\left(k+1\right)T\right]$ is the duration of non-effectiveness. θ is the killing rate of ACPs in the duration of effectiveness.

The behavior of ACPs is governed by the parameters $p_{1}$ and $q_{1}$ which refer to the settling bias from host to weed and from weed to host, respectively, in the duration of effectiveness of the pesticide $\left(kT,\;kT+T_{1}\right]$. Further, in the duration of non-effectiveness $\left(kT+T_{1},\;\left(k+1\right)T\right]$, the parameters of the settling bias denote $p_{2}$ and $q_{2}$.

With the above assumptions, we establish a multi-host switching HLB model:(1)$$\left\{\begin{array}{cc} \frac{dS_{h}}{\mathrm{dt}} & =\mu_{h}N_{h}+\gamma I_{h}-\frac{\beta_{1}S_{h}Y_{c}}{N_{h}}-\mu_{h}S_{h},\\ \frac{dI_{h}}{\mathrm{dt}} & =\frac{\beta_{1}S_{h}Y_{c}}{N_{h}}-\mu_{h}I_{h}-\gamma I_{h},\\ \frac{dW}{\mathrm{dt}} & =\Lambda_{w}-\mu_{w}W,\\ \frac{dX_{c}}{\mathrm{dt}} & =\Lambda_{V}-\frac{\beta_{2}X_{c}I_{h}}{N_{h}}-p_{1}\delta X_{c}\frac{\alpha W}{1+\alpha_{1}W}+q_{1}\delta X_{w}-d_{c}X_{c}-\theta X_{c},\\ \frac{dY_{c}}{\mathrm{dt}} & =\frac{\beta_{2}X_{c}I_{h}}{N_{h}}-p_{1}\delta Y_{c}\frac{\alpha W}{1+\alpha_{1}W}+q_{1}\delta Y_{w}-d_{c}Y_{c}-\theta Y_{c},\\ \frac{dX_{w}}{\mathrm{dt}} & =p_{1}\delta X_{c}\frac{\alpha W}{1+\alpha_{1}W}-q_{1}\delta X_{w}-d_{w}X_{w},\\ \frac{dY_{w}}{\mathrm{dt}} & =p_{1}\delta Y_{c}\frac{\alpha W}{1+\alpha_{1}W}-q_{1}\delta Y_{w}-d_{w}Y_{w}, \end{array}\right.\;\mathrm{for}\;\;t\in \left(kT,\;kT+T_{1}\right],$$
and
(2)$$\left\{\begin{array}{cc} \frac{dS_{h}}{\mathrm{dt}} & =\mu_{h}N_{h}+\gamma I_{h}-\frac{\beta_{1}S_{h}Y_{c}}{N_{h}}-\mu_{h}S_{h},\\ \frac{dI_{h}}{\mathrm{dt}} & =\frac{\beta_{1}S_{h}Y_{c}}{N_{h}}-\mu_{h}I_{h}-\gamma I_{h},\\ \frac{dW}{\mathrm{dt}} & =\Lambda_{w}-\mu_{w}W,\\ \frac{dX_{c}}{\mathrm{dt}} & =\Lambda_{V}-\frac{\beta_{2}X_{c}I_{h}}{N_{h}}-p_{2}\delta X_{c}\frac{\alpha W}{1+\alpha_{1}W}+q_{2}\delta X_{w}-d_{c}X_{c},\\ \frac{dY_{c}}{\mathrm{dt}} & =\frac{\beta_{2}X_{c}I_{h}}{N_{h}}-p_{2}\delta Y_{c}\frac{\alpha W}{1+\alpha_{1}W}+q_{2}\delta Y_{w}-d_{c}Y_{c},\\ \frac{dX_{w}}{\mathrm{dt}} & =p_{2}\delta X_{c}\frac{\alpha W}{1+\alpha_{1}W}-q_{2}\delta X_{w}-d_{w}X_{w},\\ \frac{dY_{w}}{\mathrm{dt}} & =p_{2}\delta Y_{c}\frac{\alpha W}{1+\alpha_{1}W}-q_{2}\delta Y_{w}-d_{w}Y_{w}, \end{array}\right.\;\mathrm{for}\;\;t\in \left(kT+T_{1},\;\left(k+1\right)T\right],$$
where $\mu_{h}$ denotes the natural death rate of citrus trees, γ denotes the rouging rate of citrus trees, $\mu_{w}$ is the mortality rate of *A. conyzoides*, and $d_{c}$ and $d_{w}$ are the natural death rate of ACPs in the trees and *A. conyzoides*, respectively. All parameters and its biological interpretation of model (Equation 1) and (Equation 2) are summarized in Table 1.

It follows from the third equation of model (Equation 1) and (Equation 2) that
$$\lim_{t\to \infty }W\left(t\right)=\frac{\Lambda_{w}}{\mu_{w}}≐W^{*}.$$
This allows us to solve system (Equation 1) and (Equation 2) by studying the limit system:(3)$$\left\{\begin{array}{cc} \frac{dS_{h}}{\mathrm{dt}} & =\mu_{h}K+\gamma I_{h}-\frac{\beta_{1}S_{h}Y_{c}}{K}-\mu_{h}S_{h},\\ \frac{dI_{h}}{\mathrm{dt}} & =\frac{\beta_{1}S_{h}Y_{c}}{K}-\mu_{h}I_{h}-\gamma I_{h},\\ \frac{dX_{c}}{\mathrm{dt}} & =\Lambda_{V}-\frac{\beta_{2}X_{c}I_{h}}{K}-p_{1}\delta X_{c}\frac{\alpha W^{*}}{1+\alpha_{1}W^{*}}+q_{1}\delta X_{w}-d_{c}X_{c}-\theta X_{c},\\ \frac{dY_{c}}{\mathrm{dt}} & =\frac{\beta_{2}X_{c}I_{h}}{K}-p_{1}\delta Y_{c}\frac{\alpha W^{*}}{1+\alpha_{1}W^{*}}+q_{1}\delta Y_{w}-d_{c}Y_{c}-\theta Y_{c},\\ \frac{dX_{w}}{\mathrm{dt}} & =p_{1}\delta X_{c}\frac{\alpha W^{*}}{1+\alpha_{1}W^{*}}-q_{1}\delta X_{w}-d_{w}X_{w},\\ \frac{dY_{w}}{\mathrm{dt}} & =p_{1}\delta Y_{c}\frac{\alpha W^{*}}{1+\alpha_{1}W^{*}}-q_{1}\delta Y_{w}-d_{w}Y_{w}, \end{array}\right.\;\mathrm{for}\;\;t\in \left(kT,\;kT+T_{1}\right],$$
and
(4)$$\left\{\begin{array}{cc} \frac{dS_{h}}{\mathrm{dt}} & =\mu_{h}K+\gamma I_{h}-\frac{\beta_{1}S_{h}Y_{c}}{K}-\mu_{h}S_{h},\\ \frac{dI_{h}}{\mathrm{dt}} & =\frac{\beta_{1}S_{h}Y_{c}}{K}-\mu_{h}I_{h}-\gamma I_{h},\\ \frac{dX_{c}}{\mathrm{dt}} & =\Lambda_{V}-\frac{\beta_{2}X_{c}I_{h}}{K}-p_{2}\delta X_{c}\frac{\alpha W^{*}}{1+\alpha_{1}W^{*}}+q_{2}\delta X_{w}-d_{c}X_{c},\\ \frac{dY_{c}}{\mathrm{dt}} & =\frac{\beta_{2}X_{c}I_{h}}{K}-p_{2}\delta Y_{c}\frac{\alpha W^{*}}{1+\alpha_{1}W^{*}}+q_{2}\delta Y_{w}-d_{c}Y_{c},\\ \frac{dX_{w}}{\mathrm{dt}} & =p_{2}\delta X_{c}\frac{\alpha W^{*}}{1+\alpha_{1}W^{*}}-q_{2}\delta X_{w}-d_{w}X_{w},\\ \frac{dY_{w}}{\mathrm{dt}} & =p_{2}\delta Y_{c}\frac{\alpha W^{*}}{1+\alpha_{1}W^{*}}-q_{2}\delta Y_{w}-d_{w}Y_{w}, \end{array}\right.\;\mathrm{for}\;\;t\in \left(kT+T_{1},\;\left(k+1\right)T\right],$$
with initial conditions
(5)$$S_{h}\left(0\right)>0,I_{h}\left(0\right)\geq 0,X_{c}\left(0\right)\geq 0,Y_{c}\left(0\right)\geq 0,X_{w}\left(0\right)\geq 0,Y_{w}\left(0\right)\geq 0.$$


### 2.2. Model Parameters

The parameter estimation of the model is crucial for the study of epidemics. Here, 11 parameters are set to realistic values found in the literature. However, due to a lack of data on the vector bias for plants in the *Asteraceae* and *non-Asteraceae* families, certain parameters in Table 2 are assigned assumed values. It is important to note that during the period when the pesticide is effective, some ACPs spread from citrus trees to *A. conyzoides*, while ACPs on *A. conyzoides* hardly spread to citrus trees. However, during the period when the pesticide is ineffective, the situation is reversed. As a result, we assume the preference parameters for plants in the Asteraceae family (citrus trees) and non-Asteraceae fmily (*A. conyzoides*) are $p_{2}=0$ and $q_{1}=0$.

## 3. Analytical Results

Prior to delving the analysis of system (Equation 3) and (Equation 4), it is necessary to introduce some notations and establish key finndings for the linear switching system in a periodic environment. Define $R_{+}=\left\{x\in R\;|\;x\geq 0\right\}$, $R_{+}^{n}=\left\{x\in R^{n}\;|\;x_{i}\geq 0,i=1,2,...,n\right\}$. Let r(B) be the spectral radius of matrix *B*.

### 3.1. Some Results for Linear Switching System

Consider the following linear switching periodic system:(6)$$\frac{dx(t)}{dt}=A_{k}x\left(t\right),\;\;\;t\in \left(t_{k-1},\;t_{k}\right],$$
where $x=\left(x_{1},\;x_{2},\;\cdots ,\;x_{n}\right)\in R^{n}$, $A_{k}\in R^{n\times n}$, *q* is a fixed positive integer such that $A_{k+q}=A_{k}$, $t_{k}-t_{k-1}=T_{k}$ with $T_{k+q}=T_{k}$, and then $T=\sum_{k=1}^{q}T_{k}$ is the period of switch system.

Denote
(7)$$\Phi_{A_{k}}\left(T\right):=\prod_{k=1}^{q}\exp\left(A_{q-k+1}T_{q-k+1}\right).$$

It is important to note that system (Equation 6) can be considered the special case of system (5) in [22]. While there is no pulse present, it degenerates to the system (Equation 6) in this paper. According to Lemma 1 in [22], we have the following results.

**Lemma 1.** 
*If $\eta =\left(1/T\right)\ln r(\Phi_{A_{k}}\left(T\right)),$ then there exists a positive T-periodic vector function ν(t) such that exp(ηt)ν(t) is a solution of the linear T-periodic switching system (Equation 6).*


**Lemma 2.** 
*If $r(\Phi_{A_{k}}\left(T\right))<1,$ then the trivial solution of system (Equation 6) is asymptotically stable.*


### 3.2. Basic Reproduction Number for General Periodic Switching System

The basic reproduction number, $R_{0}$, is the number of newly infected plants that arise from one infected plant in a whole susceptible plant population [32]. In the last few decades, $R_{0}$ has become a fundamental parameter in mathematical epidemiology and has been widely applied in the study of the dynamics of animal and plant epidemics [33,34]. In classical epidemic models, the basic reproduction number serves as a threshold determinant. It is a common case that a disease dies out if the basic reproduction number, $R_{0}$, is less than 1, and the disease persists whenever $R_{0}$ is greater than 1.

For autonomous continuous-time epidemic models, the calculation of the basic reproduction number is typically performed using the next-generation matrix method, introduced by van den Driessche and Watmough [35]. However, for non-autonomous systems [36], impulsive systems [37], and impulsive and switching systems [22], corresponding explicit formulae have been developed to calculate the basic reproduction number using the linear operator method.

To calculate the basic reproduction number for the switching system (Equation 3) and (Equation 4), it is necessary to first examine a general switching system in a periodic environment:(8)$$\frac{dx(t)}{dt}=f^{k}\left(x\right),\;\;\;\mathrm{for}\;\;t\in \left(t_{k-1},t_{k}\right],$$
where $f^{k}:R_{+}^{n}\to R^{n}$, $f^{k+q}=f^{k}$, $t_{k}-t_{k-1}=T_{k}$ with $T_{k+q}=T_{k}$, and then $T=\sum_{k=1}^{q}T_{k}$ is the period of the switch system. Note that system (Equation 8) is the special case of system (9) in [22].

The basic reproduction number is derived by following the linear operator method as presented in [22]. Following the notation from Gao et al. [22], the first *m* compartments $x_{1},x_{2},\cdots ,x_{m}$ denote the infected individuals; $x_{m+1},x_{2},\cdots ,x_{n}$ the uninfected individuals; $X_{s}$ represents the set of all disease-free state, i.e., $X_{s}=\left\{x\in R_{+}^{n}|x_{i}=0,i=1,\cdots ,m\right\}$; and $X=(x_{1},x_{2},\cdots ,x_{m})$, $Y=(x_{m+1},x_{2},\cdots ,x_{n}).$

We can rewrite system (Equation 8) as:(9)$$\begin{array}{c} \frac{dx(t)}{dt}=F^{k}\left(x\left(t\right)\right)-V^{k}\left(x\left(t\right)\right),\;\;\;\mathrm{for}\;\;t\in \left(t_{k-1,t_{k}}\right]. \end{array}$$
where $F^{k}\left(x\right)$ are the newly infected rates, $V^{k}\left(x\right)=V^{k-}\left(x\right)-V^{k+}\left(x\right)$ represent the set transfer rates out of compartments, here $V^{k+}\left(x\right)$ are the input rates of individuals by other means, and $V^{k-}\left(x\right)$ are the rates of transfer of individuals out of compartments. Thus, $f^{k}\left(x\right)=F^{k}\left(x\right)-V^{k}\left(x\right).$ We assume that system (Equation 9) has a disease-free periodic solution $x^{*}\left(t\right).$

Denote
(10)$$\begin{array}{c} F_{k}\left(t\right)={\left(\frac{\partial F_{i}^{k}\left(x^{*}\left(t\right)\right)}{\partial x_{j}}\right)}_{1\leq i,j\leq m}\;\;\mathrm{and}\;\;V_{k}\left(t\right)={\left(\frac{\partial F_{i}^{k}\left(x^{*}\left(t\right)\right)}{\partial x_{j}}\right)}_{1\leq i,j\leq m}. \end{array}$$

We make the following assumptions, which share the same biological meanings as those by Gao et al. [22].

**Hypothesis 1.** 
*If $x_{i}\geq 0$, then the function $F_{i}^{k}\left(x\right)$, $V_{i}^{k-}\left(x\right)$ and $V_{i}^{k+}\left(x\right)$ are nonnegative and continuous on $R_{+}^{n}$ and continuously differential with respect to x for i=1,⋯,n.*


**Hypothesis 2.** 
*If $x_{i}=0$, then $V_{i}^{k-}\left(x\right)=0$. Particularly, if $x\in X_{s}$, then $V_{i}^{k-}\left(x\right)=0$ for i=1,⋯,m.*


**Hypothesis 3.** 
*$F_{i}^{k}\left(x\right)=0$ for i=m+1,⋯,n.*


**Hypothesis 4.** 
*If $x\in X_{s}$, then $F_{i}^{k}\left(x\right)=V_{i}^{k+}\left(x\right)=0$ for i=1,⋯,m.*


**Hypothesis 5.** 
*$r(\Phi_{M_{k}}\left(T\right))<1$, where $\Phi_{M_{k}}\left(T\right)=\prod_{k=1}^{q}\exp\left(M_{q-k+1}T_{q-k+1}\right)$, and $\Phi_{M_{k}}\left(t\right)$ is the fundamental solution matrix of the following system:*

$$\begin{array}{c} \frac{dz(t)}{dt}=M_{k}\left(t\right)z\left(t\right), \end{array}$$

*where*

(11)
$$\begin{array}{c} M_{k}\left(t\right)={\left(\frac{\partial f_{i}^{k}\left(x^{*}\left(t\right)\right)}{\partial x_{j}}\right)}_{m+1\leq i,j\leq n}. \end{array}$$



**Hypothesis 6.** 

$r(\Phi_{-V_{k}}\left(T\right))<1.$


*Further, let Y(t,s)(t>s) be the evolution operator of the following linear switching system:*

(12)
$$\begin{array}{c} \frac{dy(t)}{dt}=-V_{k}\left(t\right)y\left(t\right),\;for\;t\in \left(t_{k-1},t_{k}\right]. \end{array}$$

*Similar to the notation and definition of [22], we define the so-called next infection operator L,*

(13)
$$\begin{array}{ccc} L\phi (t) & =\int_{-\infty }^{t}Y\left(t,s\right)F\left(s\right)\phi \left(s\right)ds\; & =\int_{0}^{+\infty }Y\left(t,t-a\right)F\left(t-a\right)\phi \left(t-a\right)da,\;\forall t\in R_{+}, \end{array}$$

*where ϕ(s) is a T-periodic function from R to $R_{+}^{m}$ and denotes the initial distribution of infections individuals, and $F\left(t\right)=F_{k}\left(t\right)$ for $t\in (t_{k-1},\;t_{k}]$. Now, we define the basic reproduction number $R_{0}$ for system (Equation 9) as:*

(14)
$$\begin{array}{c} R_{0}=r\left(L\right). \end{array}$$

*In order to calculate the implicit expression $R_{0}$ by numerical simulation, we consider the auxiliary T-periodic switching system:*

(15)
$$\begin{array}{c} \frac{dU(t)}{dt}=\left(-V_{k}\left(t\right)+\frac{F_{k}\left(t\right)}{\lambda }\right)U\left(t\right). \end{array}$$

*where λ∈(0,∞). Set U(t,s,λ)(t≥s) to be the evolution operator of system (Equation 15), then $U\left(T,0,\lambda \right)=\Phi_{\left(F_{k}/\lambda \right)-V_{k}}\left(T\right)$. According to Lemmas 3 and 4 of [22], the following results can be yielded.*


**Lemma 3.** 
*Assuming that (*
*
**H1**
*
*)–(*
*
**H6**
*
*) hold, then the following statements are valid:*
*(i)* 
*If $r(\Phi_{\left(F_{k}/\lambda \right)-V_{k}}\left(T\right))=1$ has a positive solution $\lambda_{0},$ then $\lambda_{0}$ is an eigenvalue of L, and so $R_{0}>0$.*
*(ii)* 
*If $R_{0}>0$, then $\lambda =R_{0}$ is the unique solution of $r(\Phi_{\left(F_{k}/\lambda \right)-V_{k}}\left(T\right))=1$.*
*(iii)* 
*$R_{0}=0$ if and only if $r(\Phi_{\left(F_{k}/\lambda \right)-V_{k}}\left(T\right))<1$ for all λ>0.*



In view of the results of Lemma 3, we have that $R_{0}$ for the periodic switching system (Equation 8) is the solution of algebraic equation $r(\Phi_{\left(\left(F_{k}/\lambda \right)-V_{k}\right)}\left(T\right))=1$.

**Lemma 4.** 
*Assuming that (*
*
**H1**
*
*)–(*
*
**H6**
*
*) hold, then the following statements are valid for system (Equation 9):*
*(i)* 
*$R_{0}=1$ if and only if $r(\Phi_{\left(F_{k}-V_{k}\right)}\left(T\right))=1$.*
*(ii)* 
*$R_{0}>1$ if and only if $r(\Phi_{\left(F_{k}-V_{k}\right)}\left(T\right))>1$.*
*(iii)* 
*$R_{0}<1$ if and only if $r(\Phi_{\left(F_{k}-V_{k}\right)}\left(T\right))<1$.*



It follows from Lemma 4 that the disease-free periodic solution $x^{*}\left(t\right)$ of system (Equation 9) is asymptotically stable if $R_{0}<1$ and unstable if $R_{0}>1$.

To proof our main result, we state the Spectral Mapping Theorem (see Theorem 1.4 in [38]) which will be essential to our proof.

**Lemma 5.** 
*Let g(t) be a polynomial with complex coefficients, and let the eigenvalues of n×n matrix A be $\lambda_{1},\;\lambda_{2},\cdots ,\;\lambda_{n}$. Then, the eigenvalues of g(A) are $f\left(\lambda_{1}\right),\;f\left(\lambda_{2}\right),\cdots ,f\left(\lambda_{n}\right)$.*


### 3.3. Dynamics of Switching Model (Equation 3) and (Equation 4)

#### 3.3.1. Non-Negativity and Boundedness

Let
$$\Omega =\left\{\left(S_{h},I_{h},X_{c},Y_{c},X_{w},Y_{w}\right)\in R_{+}^{6}\;|\;S_{h}+I_{h}=K,\;X_{c}+Y_{c}+X_{w}+Y_{w}\leq \frac{\Lambda_{v}}{d_{\min}}\right\},$$
where $d_{\min}=\min\left\{d_{c},d_{w},\theta \right\}$. In the following, we will show that switching system (Equation 3) and (Equation 4) is well posed in Ω.

**Lemma 6.** *The feasible region* Ω *is positively invariant and attracts all solutions of system (Equation 3) and (Equation 4).*

**Proof.** Let $\xi \left(t\right)=(S_{h}\left(t\right),\;I_{h}\left(t\right),\;X_{h}\left(t\right),\;Y_{h}\left(t\right),\;X_{w}\left(t\right),\;Y_{w}\left(t\right))$ be any solution of switching system (Equation 3) and (Equation 4) with initial conditions (Equation 5). We first show the non-negativity of solutions. Set $t_{1}=\sup\left\{t>0\;|\;\xi \left(s\right)>0,\;\mathrm{for}\;s\in \left[0,\;t\right)\right\}$. Obviously, $t_{1}>0$. It follows from the first equation of (Equation 3) and (Equation 4) that
(16)$$\frac{dS_{h}}{\mathrm{dt}}=\mu_{h}K+\gamma I_{h}-\frac{\beta_{1}S_{h}Y_{c}}{K}-\mu_{h}S_{h}.$$Denote $\lambda_{h}\left(t\right)=\frac{\beta_{1}Y_{c}}{K}$, then (Equation 16) becomes
$$\frac{dS_{h}}{\mathrm{dt}}=\mu_{h}K+\gamma I_{h}-\lambda_{h}\left(t\right)S_{h}-\mu_{h}S_{h},$$
which can be re-written as
$$\frac{d}{dt}\left\{S_{h}\left(t\right)\exp\left(\int_{0}^{t}\lambda_{h}\left(s\right)ds+\mu_{h}t\right)\right\}=\left(\mu_{h}K+\gamma I_{h}\right)\exp\left(\int_{0}^{t}\lambda_{h}\left(s\right)ds+\mu_{h}t\right).$$Thus,
$$\begin{array}{cc} \; & S_{h}\left(t_{1}\right)\exp\left(\int_{0}^{t_{1}}\lambda_{h}\left(s\right)ds+\mu_{h}t_{1}\right)-S_{h}\left(0\right)\\ = & \int_{0}^{t_{1}}\left(\mu_{h}K+\gamma I_{h}\left(\zeta \right)\right)\cdot \exp\left(\int_{0}^{\zeta }\lambda_{h}\left(s\right)ds+\mu_{h}\zeta \right)d\zeta . \end{array}$$Consequently,
$$\begin{array}{cc} S_{h}\left(t_{1}\right)= & S_{h}\left(0\right)\exp\left(-\int_{0}^{t_{1}}\lambda_{h}\left(s\right)ds+\mu_{h}t_{1}\right)\\ \; & +\exp\left(-\int_{0}^{t_{1}}\lambda_{h}\left(s\right)ds+\mu_{h}t_{1}\right)\int_{0}^{t_{1}}\left(\mu_{h}K+\gamma I_{h}\left(\zeta \right)\right)\cdot \exp\left(\int_{0}^{\zeta }\lambda_{h}\left(s\right)ds+\mu_{h}\zeta \right)d\zeta \\ > & 0. \end{array}$$Similarly, it can be proven that ξ(t)≥0 for all t>0.Next, we need to show the boundedness of the solutions. Set $N_{v}=X_{c}+Y_{c}+X_{w}+Y_{w}$. Adding the last four equations of (Equation 3) and (Equation 4), we have
$$\frac{dN_{v}}{\mathrm{dt}}\leq \Lambda_{h}-d_{min}N_{v},$$
which implies that $N_{v}\leq \frac{\Lambda_{h}}{d_{min}}$ for all t≥0. Therefore, the region Ω is positively invariant with respect to the switching system (Equation 3) and (Equation 4). □

The results of Lemma 6 show that it is sufficient to study the dynamic properties of the switching system (Equation 3) and (Equation 4) in Ω, which we present in the following subsections.

#### 3.3.2. Threshold Dynamics

In this subsection, we will explore the threshold condition which leads to the extinction and persistence of the disease for the switching system (Equation 3) and (Equation 4).

Note that the switching system (Equation 3) and (Equation 4) is the special case of the general switching system (Equation 8), in which $x={\left(I_{h},Y_{c},Y_{w},S_{h},W,X_{c},X_{w}\right)}^{T}$, q=2, $t_{2k}=kT,t_{2k+1}=kT+T_{1}$, $f^{k+2}\left(x\right)=f^{k}\left(x\right)$. Thus,
(17)$$f^{2k}\left(x\right)=\left(\begin{array}{c} \frac{\beta_{1}S_{h}Y_{c}}{K}-\mu_{h}I_{h}-\gamma I_{h}\\ \frac{\beta_{2}X_{c}I_{h}}{K}-p_{1}\delta Y_{c}\frac{\alpha W^{*}}{1+\alpha_{1}W^{*}}+q_{1}\delta Y_{w}-d_{c}Y_{c}-\theta Y_{c}\\ p_{1}\delta Y_{c}\frac{\alpha W^{*}}{1+\alpha_{1}W^{*}}-q_{1}\delta Y_{w}-d_{w}Y_{w}\\ \mu_{h}K+\gamma I_{h}-\frac{\beta_{1}S_{h}Y_{c}}{K}-\mu_{h}S_{h}\\ \Lambda_{V}-\frac{\beta_{2}X_{c}I_{h}}{K}-p_{1}\delta X_{c}\frac{\alpha W^{*}}{1+\alpha_{1}W^{*}}+q_{1}\delta X_{w}-d_{c}X_{c}-\theta X_{c}\\ p_{1}\delta X_{c}\frac{\alpha W^{*}}{1+\alpha_{1}W^{*}}-q_{1}\delta X_{w}-d_{w}X_{w} \end{array}\right)$$
and
(18)$$f^{2k+1}\left(x\right)=\left(\begin{array}{c} \frac{\beta_{1}S_{h}Y_{c}}{K}-\mu_{h}I_{h}-\gamma I_{h}\\ \frac{\beta_{2}X_{c}I_{h}}{K}-p_{2}\delta Y_{c}\frac{\alpha W^{*}}{1+\alpha_{1}W^{*}}+q_{2}\delta Y_{w}-d_{c}Y_{c}\\ p_{2}\delta Y_{c}\frac{\alpha W^{*}}{1+\alpha_{1}W^{*}}-q_{2}\delta Y_{w}-d_{w}Y_{w}\\ \mu_{h}K+\gamma I_{h}-\frac{\beta_{1}S_{h}Y_{c}}{K}-\mu_{h}S_{h}\\ \Lambda_{V}-\frac{\beta_{2}X_{c}I_{h}}{K}-p_{2}\delta X_{c}\frac{\alpha W^{*}}{1+\alpha_{1}W^{*}}+q_{2}\delta X_{w}-d_{c}X_{c}\\ p_{2}\delta X_{c}\frac{\alpha W^{*}}{1+\alpha_{1}W^{*}}-q_{2}\delta X_{w}-d_{w}X_{w} \end{array}\right)$$

It is easy to see that the switch system (Equation 3) and (Equation 4) has a unique disease-free periodic solution $x^{*}\left(t\right)=\left(K,\;X_{c}^{*}\left(t\right),\;0,\;X_{w}^{*}\left(t\right),\;0\right)$.

By (Equation 10) and (Equation 11), we can calculate $F_{k}$, $V_{2k}$, $V_{2k+1}$, $M_{2k}$, and $M_{2k+1}$ of the switch system (Equation 3) and (Equation 4), which are represented as the following form:$$F_{k}=\left(\begin{array}{ccc} 0 & \beta_{1} & 0\\ \beta_{2} & 0 & 0\\ 0 & 0 & 0 \end{array}\right),\;V_{2k}=\left(\begin{array}{ccc} \mu_{h}+\gamma & 0 & 0\\ 0 & d_{c}+\theta +p_{1}\delta \frac{\alpha W^{*}}{1+\alpha_{1}W^{*}} & -q_{1}\delta \\ 0 & -p_{1}\delta \frac{\alpha W^{*}}{1+\alpha_{1}W^{*}} & d_{w}+q_{1}\delta \end{array}\right),$$
$$V_{2k+1}=\left(\begin{array}{ccc} \mu_{h}+\gamma & 0 & 0\\ 0 & d_{c}+p_{2}\delta \frac{\alpha W^{*}}{1+\alpha_{1}W^{*}} & -q_{2}\delta \\ 0 & -p_{2}\delta \frac{\alpha W^{*}}{1+\alpha_{1}W^{*}} & d_{w}+q_{2}\delta \end{array}\right),$$
$$M_{2k}=\left(\begin{array}{ccc} -\mu_{1} & 0 & 0\\ 0 & -p_{1}\delta \frac{\alpha W^{*}}{1+\alpha_{1}W^{*}}-d_{c}-\theta & q_{1}\delta \\ 0 & p_{1}\delta \frac{\alpha W^{*}}{1+\alpha_{1}W^{*}} & -q_{1}\delta -d_{w} \end{array}\right),$$
$$M_{2k+1}=\left(\begin{array}{ccc} -\mu_{1} & 0 & 0\\ 0 & -p_{2}\delta \frac{\alpha W^{*}}{1+\alpha_{1}W^{*}}-d_{c} & q_{2}\delta \\ 0 & p_{2}\delta \frac{\alpha W^{*}}{1+\alpha_{1}W^{*}} & -q_{2}\delta -d_{w} \end{array}\right).$$

In order to derive the basic reproductive number of system (Equation 3) and (Equation 4), we need to show that Assumptions (**H1**)–(**H6**) hold. The mathematical details can be found in Appendix A.

**Theorem 1.** 
*If $R_{0}<1$, then the disease-free periodic solution $x^{*}\left(t\right)$ of system (Equation 3) and (Equation 4) is globally asymptotically stable, whereas it is unstable if $R_{0}>1$.*


The proof of Theorem 1 is shown in Appendix B. Similar to the proof of Theorem 4.1 of [22], we can obtain the uniform persistence of system (Equation 3) and (Equation 4).

**Theorem 2.** 
*If $R_{0}>1$, then the disease is uniformly persistent for system (Equation 3) and (Equation 4), that is, there is a positive constant ϵ>0, such that $\lim\inf_{t\to \infty }I_{h}\left(t\right)>\epsilon ,$ $\lim\inf_{t\to \infty }Y_{c}\left(t\right)>\epsilon ,and$ $\lim\inf_{t\to \infty }Y_{w}\left(t\right)>\epsilon .$*


Theorems 1 and 2 demonstrate that $R_{0}$ is a sharp threshold value which determines whether the disease dies out or not. If $R_{0}<1$, then the disease will be controlled, whereas if $R_{0}>1$, the disease will be endemic.

## 4. Numerical Simulation

In this section, we present numerical simulations of the system (Equation 3) and (Equation 4) to support our analytical results, and determine the optimal number of *A. conyzoides* retained in the orchard and the best period of pesticide spraying.

### 4.1. Theory Verification

Figure 2a–f are the time dynamics of the compartmental population ($S_{h}$, $I_{h}$, $X_{c}$$Y_{c}$, $X_{w}$ and $Y_{w}$) with $\Lambda_{w}=1$ and $\Lambda_{w}=9$. If $\Lambda_{w}=1$, then the basic reproduction number takes the valve $R_{0}=0.9432<1$ by numerical computation. According to Theorem 1, the disease-free periodic solution of the switching system (Equation 3) and (Equation 4) is globally asymptotically stable. Thus, the disease will die out. Further, if we fix $\Lambda_{w}=9$, then $R_{0}=1.0286>1$. By Theorem 2, we know that the disease is uniformly permanent. A numerical simulation of the above results can be seen in Figure 2.

### 4.2. Sensitive Analysis

Note that several fundamental parameters including the recruitment of *A. conyzoides* ($\Lambda_{w}$), the preference parameters ($p_{1}$, $q_{2}$), and the timing of pesticide application ($T_{1},T_{2}$) play a significant role in our model. By considering pulse parameters, we are able to investigate the quantity of weed introductions, the settling preference of ACPs, and the pesticide spraying period affecting the transmission of the disease. These factors are crucial in understanding the dynamics and control strategy of the disease within the context of our model.

To illustrate that the evolution of disease transmission evolves with increasing numbers of *A. conyzoides*, we have plotted the basic reproduction numbers in Figure 3 for different infection rates ($\beta_{1},\;\beta_{2}$), which reveal some important issues related to HLB outbreaks. When the infection rates ($\beta_{1}$ and $\beta_{2}$) are low, the basic reproduction number ($R_{0}$) monotonically increases with the parameter $\Lambda_{w}$ (see Figure 3a). However, when the infection rates reach certain values, the basic reproduction number first decreases monotonically with parameter $\Lambda_{w}$ and then increases monotonically (see Figure 3b–d). We can observe that $R_{0}$ is more sensitive when the parameter $\Lambda_{w}$ is smaller.

Next, we examine the responses of the basic reproduction number when pairs of vector preference parameters are simultaneously altered. Figure 4 illustrates that the value of $R_{0}$ increases as the landing preference parameter $p_{1}$ increases or as $q_{2}$ decreases. The threshold in Figure 3a represents the combination of landing preference parameters between citrus trees and weeds at which $R_{0}$ equals one.

From the plot, we can observe that when $p_{1}$ is less than 1.633, $R_{0}$ remains below one for all values of $q_{2}$. Conversely, when $p_{1}$ exceeds 6.122, $R_{0}$ surpasses one for $q_{2}$ values up to 20. This implies that during the period of pesticide effectiveness, if only a small proportion of Asian citrus psyllids diffuse from citrus trees to weeds, the spread of Huanglongbing can be controlled. However, if the proportion is large enough, even with a high diffusion preference parameter from weeds to citrus trees, the disease will persist. Therefore, the landing preference parameter $p_{1}$ from citrus trees to weeds plays a crucial role in the control of Huanglongbing.

Figure 5 displays the paired effects of simultaneously varying $T_{1}$ and $T_{2}$ on the basic reproduction number $R_{0}$, with a fixed value of $\beta_{2}=0.00226$. As $T_{2}$ increases, $R_{0}$ also increases. Conversely, as $T_{1}$ increases, $R_{0}$ decreases. The red line represents the threshold value where $R_{0}$ equals 1, while the different colors indicate increasing values of $R_{0}$ from 0 to 3 in increments of 0.5 (ranging from blue to yellow).

We observe that there is a rapid increase in $R_{0}$ as $T_{2}$ increases for very small values of $T_{1}$. However, for larger values of $T_{1}$, there is little change in $R_{0}$ as $T_{2}$ increases. Field experiments have indicated that when spraying 1.8% Avermectin EC at a concentration of 90 mg/L on citrus branches, the pesticide has a persistent duration of 11 days [10]. From Figure 5, we can deduce that if $T_{1}$ is set to 11 and $T_{2}$ is less than 18, then $R_{0}$ remains below 1. Therefore, to effectively control the spread of the disease, the period of spraying Avermectin should be extended to 29 days.

In Figure 6, violin plots were employed to visualize the distribution and probability density of the total number of the ACP population under different recruitment rates and infection rates. Violin plots combine the features of density plots and box plots, providing information about the median, quartiles, outliers, and density distribution of the data.

From the violin plots in Figure 6a,b, it is evident that the median values of the respective data distributions increase with higher recruitment rates for *Ageratum conyzoides* ($\Lambda_{w}$). This indicates that increasing the recruitment rate of the weed has a greater impact on the number of ACPs.

However, when considering the influence of outliers, it can be observed that the infection rates of the disease have a minimal impact on the population of ACPs. This suggests that the variability in the number of the ACP population due to infection rates is relatively small compared to the influence of other factors such as weed recruitment rates.

In Figure 6a, when the infection rate is low, comparing the medians reveals that the rate of change in the number of ACPs increases with a higher recruitment rate of *Ageratum conyzoides* ($\Lambda_{w}$). However, in Figure 6b, when the infection rate is high, the rate of change in the number of ACPs decreases with a higher recruitment rate.

Furthermore, when comparing the two violin plots, it can be observed that when the recruitment rate is low ($\Lambda_{w}=3$), the median value of ACPs is higher with high infection rates compared to low infection rates. Conversely, when the recruitment rate is high ($\Lambda_{w}=9$), the opposite trend is observed. In this case, the median value of ACPs is higher with low infection rates compared to high infection rates. Additionally, the concentration distribution of the ACP population is more clustered around the median value when the recruitment rate is high and the infection rate is low.

In the past, farmers would extensively introduce *A. conyzoides*, a weed that produces pollen that serves as an alternative food source for natural enemy predatory mites. This practice was implemented as part of integrated pest management strategies in citrus orchards. The presence of *A. conyzoides* helped to effectively maintain continuous control on citrus pest mites.

By introducing *A. conyzoides*, farmers aimed to provide a supplementary food source for predatory mites, which are natural enemies of citrus pest mites. This approach was considered beneficial as it promoted the presence and activity of predatory mites, which in turn helped to control the population of citrus pest mites. This integrated pest management approach aimed to reduce reliance on chemical pesticides and promote a more sustainable and environmentally friendly method of pest control in citrus orchards.

*A. conyzoides*, while beneficial for ecological control of predatory mites in citrus orchards, can potentially interfere with the effectiveness of pesticide control against ACPs [10,39]. To assess the impact of *A. conyzoides* on the population of ACPs and the spread of citrus HLB under pesticide control conditions, a switching differential model was established in the study.

The study also developed the theory of the basic reproduction ratio, $R_{0}$, for a class of periodic switching systems. It was proven that $R_{0}$ serves as a threshold parameter for the stability of the disease-free periodic solution of the system. Furthermore, the theory of $R_{0}$ was applied to the switching HLB model, resulting in a threshold-type result in relation to $R_{0}$.

This threshold result provides valuable insights into the dynamics of the disease and the impact of *A. conyzoides* on the spread of HLB in the presence of pesticide control measures. By understanding the threshold value of $R_{0}$, researchers and policymakers can make informed decisions regarding disease management strategies and the role of *A. conyzoides* in controlling ACP populations and HLB spread in citrus orchards.

In this paper, we have developed the theory of the basic reproduction ratio, $R_{0}$, for a specific class of periodic switching systems. It has been demonstrated that $R_{0}$ serves as a threshold parameter for determining the stability of the disease-free periodic solution of the system described in Equations (4) and (5).

Furthermore, the theory of $R_{0}$ has been applied to a multi-host switching model for HLB. We have presented a threshold-type result in relation to $R_{0}$, and it has been proven that when $R_{0}$ is less than 1, the disease will eventually die out.

This threshold result is significant as it provides a quantitative measure for assessing the potential spread and control of HLB. By determining the critical value of $R_{0}$, researchers and policymakers can evaluate the effectiveness of disease control strategies and make informed decisions to prevent and manage the spread of HLB.

Furthermore, the numerical results obtained in our study have yielded valuable insights into the transmission dynamics of Huanglongbing (HLB) and have shed light on key factors that impact disease control measures.

One important aspect we have investigated is the influence of the quantity of weed introductions on the transmission of HLB. Our numerical findings have revealed that in scenarios with low infection rates, it is more feasible to control the spread of the disease in orchards by minimizing the presence or introduction of weeds. This suggests that reducing or eliminating weeds can be an effective strategy for disease control in such cases.

However, interestingly, our results have also shown that in scenarios with high infection rates, a moderate amount of weeds can actually be beneficial for disease control. This implies that in certain situations, the presence of weeds can play a role in suppressing the spread of HLB. These findings highlight the complex interplay between weed populations and disease dynamics, emphasizing the need for a nuanced approach to disease management strategies.

Overall, our numerical results provide valuable insights into the transmission dynamics of HLB and offer guidance on the optimal management of weeds in order to effectively control the spread of the disease in citrus orchards.

Secondly, our study has also examined the responses of the basic reproduction number ($R_{0}$) to alterations in pairs of vector preference parameters. Our findings have revealed that as the selection preference parameter ($p_{1}$) increased or the diffusion preference parameter ($q_{2}$) decreased, $R_{0}$ also increased. The threshold parameter combination between citrus trees and weeds, which results in $R_{0}$ being equal to one, played a significant role in disease control.

These results indicate that during the persistence of pesticides, if only a small proportion of Asian citrus psyllids (ACPs) diffuse from citrus trees to weeds, the spread of HLB can be controlled. However, when the proportion is large enough, the disease will become permanent, even if the diffusion preference parameter from weeds to citrus trees is high. Therefore, the landing preference parameter from citrus trees to weeds, $p_{1}$, plays a key role in HLB control. This suggests that reducing the spread of ACPs from citrus trees to weeds is beneficial for controlling the population size of ACPs and the transmission of HLB.

Based on these findings, several measures can be taken for effective control of ACP population size and HLB transmission: (i) Choose the appropriate timing for pesticide application: timely spraying of pesticides during the early stages of citrus trees being infested by ACPs can prevent the pests from spreading to weeds and avoid their transmission between citrus trees and weeds. (ii) Spray pesticides to the lower canopy: direct pesticide application to the lower parts of citrus trees can minimize pesticide contact with weeds on the ground. (iii) Implement integrated pest management strategies: in addition to pesticide application, combining other control methods such as traps and biological control can comprehensively control the spread of ACPs.

In conclusion, by choosing the right timing for pesticide application, spraying pesticides to the appropriate locations, and implementing integrated pest management strategies, the spread of ACPs from citrus trees to weeds can be effectively reduced, leading to effective control of the population size of ACPs and the transmission of HLB.

Thirdly, our study investigated the paired effects of varying $T_{1}$ and $T_{2}$ simultaneously on the basic reproduction number $R_{0}$, with fixed values of $\beta_{2}$. The results, as shown in Figure 5, indicate that as $T_{2}$ increases, $R_{0}$ also increases. Conversely, as $T_{1}$ increases, $R_{0}$ decreases. The results illustrate that for very small $T_{1}$, there is a rapid rise in $R_{0}$ as $T_{2}$ increases. However, for large $T_{1}$, there is little change in $R_{0}$ as $T_{2}$ increases. Based on our analysis, we have determined that the optimal spraying period for Avermectin is 29 days.

These findings provide important insights into the optimal timing and frequency of pesticide application for effective control of ACPs and HLB transmission. By understanding the paired effects of $T_{1}$ and $T_{2}$ on $R_{0}$, farmers and policymakers can make informed decisions regarding the timing and frequency of pesticide treatments to effectively manage ACP populations and control the spread of HLB in citrus orchards.

Fourth, in our study, we have investigated the role of the ACP as a vector for the transmission of HLB disease. ACPs transmit the HLB pathogen to citrus trees through their feeding activities, specifically by injecting the pathogen into the trees. The rate at which the psyllid bites and feeds on the trees directly affects the basic reproduction number ($R_{0}$), which represents the number of healthy trees that can be infected by each infected psyllid. Therefore, the biting rate, or infection rate, of the psyllid plays a crucial role in the transmission dynamics of HLB.

To explore this further, we have utilized violin plots to display the relationship between weed recruitment rates, infection rates, and the population growth of ACPs in citrus orchards. Our results have revealed that weeds present in citrus orchards have a significant influence on the population dynamics of ACPs. The presence of weeds can provide additional food sources and breeding grounds for the psyllids, leading to increased population sizes. On the other hand, the infection intensity, or the level of HLB disease in the psyllid population, has a minimal effect on the overall population size of the psyllids.

These findings highlight the importance of considering the role of weeds in citrus orchards when developing strategies for controlling ACP populations and managing the spread of HLB. Efforts to control weeds and minimize their presence in orchards can help reduce the availability of food and breeding sites for ACPs, ultimately leading to a decrease in their population sizes. This, in turn, can contribute to the control of HLB transmission.

Overall, our study emphasizes the significance of understanding the relationship between weed recruitment rates, infection rates, and the population growth of ACPs in citrus orchards. By considering these factors, researchers and policymakers can develop targeted and effective strategies for managing ACP populations and controlling the spread of HLB disease.

## 5. Conclusions

Our study focuses on the theory of the basic reproduction number, $R_{0}$, for periodic switching systems and its application to the switching HLB model. It is proven that $R_{0}$ serves as a threshold parameter for the stability of the disease-free periodic solution of the system. The threshold result provides insights into the dynamics of the disease and the impact of *A. conyzoides* on the spread of HLB in the presence of pesticide control measures. Understanding the threshold value of $R_{0}$ helps in making informed decisions regarding disease management strategies and the role of *A. conyzoides* in controlling ACP populations and HLB spread in citrus orchards. The study also investigates the influence of weed introductions on the transmission of HLB and finds that reducing or eliminating weeds can be an effective strategy for disease control in scenarios with low infection rates.

In summary, the study highlights the interplay between recruitment rates, infection rates, and their impact on the basic reproduction number of the disease. It further emphasizes the importance of carefully considering the duration and period of pesticide application to effectively control disease spread, while also shedding light on the relationship between weed recruitment rates, infection rates, and the population growth of ACPs in citrus orchards. To achieve effective control of ACPs and HLB, the following measures should be considered: (i) Choose the appropriate timing for pesticide application: timely spraying of pesticides during the early stages of citrus trees being infested by citrus psyllids can prevent the pests from spreading to weeds and avoid their transmission between citrus trees and weeds. (ii) Spray pesticides to the lower canopy: spray pesticides to the lower parts of citrus trees, minimizing pesticide contact with weeds on the ground. (iii) Implement integrated pest management strategies: in addition to pesticide application, other control methods such as traps and biological control should also be combined to comprehensively control the spread of ACPs.

## Figures and Tables

**Figure 1 plants-12-03659-f001:**
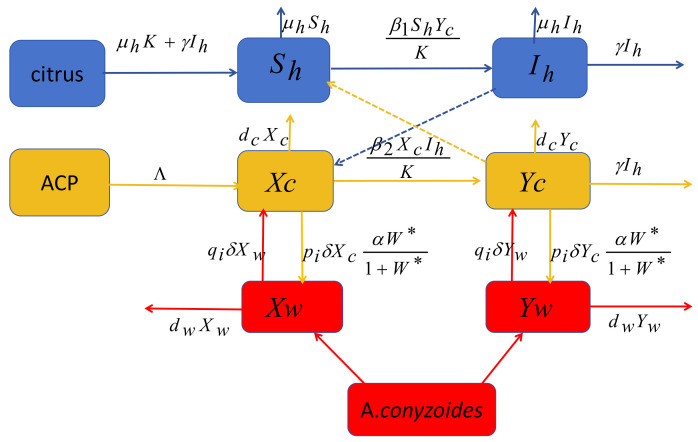
Schematic diagram of the modeling interaction of HLB transmission in citrus trees, *A. conyzoides*, and ACP populations. Trees are either susceptible or infected. Adult psyllids are either susceptible or infected. Blue, yellow and red arrows show the transitions between compartments. Blue and yellow dashed arrows show the necessary interactions between trees and psyllids to obtain transmission.

**Figure 2 plants-12-03659-f002:**
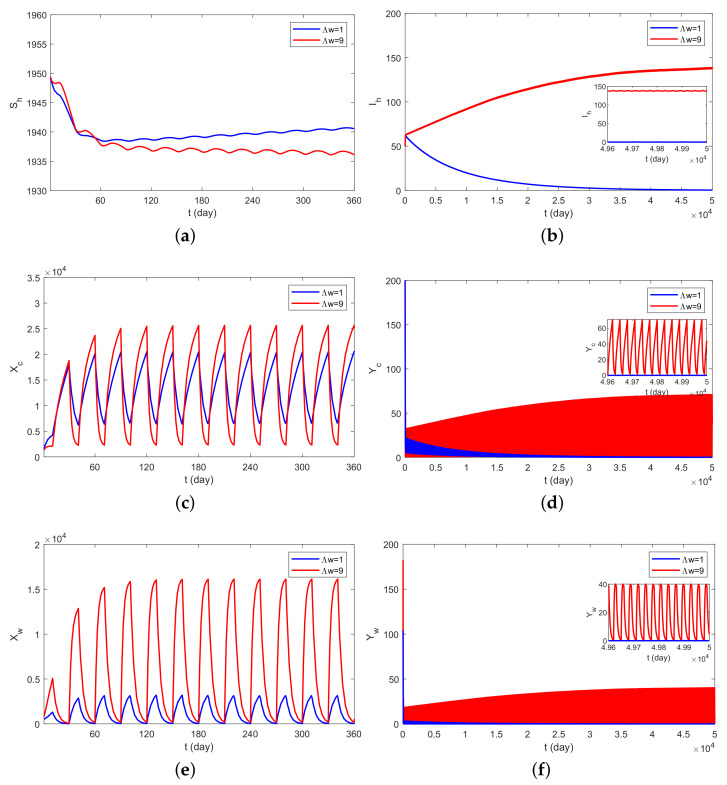
Time series of solutions for switching system (Equation 3) and (Equation 4), (**a**) $S_{h}$, (**b**) $I_{h}$, (**c**) $X_{c}$, (**d**) $Y_{c}$, (**e**) $X_{w}$, (**f**) $Y_{w}$ with $\Lambda_{w}=1$ and $\Lambda_{w}=9$, showing the disease will be extinct eventually when $R_{0}=0.9432$ (blue), and the disease is permanent when $R_{0}=1.0286$ (red).

**Figure 3 plants-12-03659-f003:**
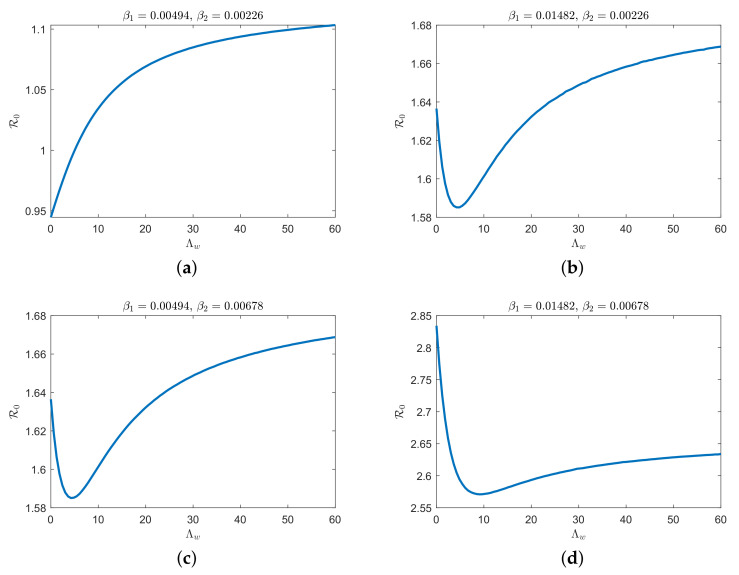
The basic reproduction number vs. changes in the constant recruitment rate for *A. conyzoides* $\Lambda_{w}$ and different infection rates, (**a**) $\beta_{1}=0.00494,\;\beta_{2}=0.00226$, (**b**) $\beta_{1}=0.01482,\;\beta_{2}=0.00226$, (**c**) $\beta_{1}=0.00494,\;\beta_{2}=0.00678$, (**d**) $\beta_{1}=0.01482,\;\beta_{2}=0.00678$.

**Figure 4 plants-12-03659-f004:**
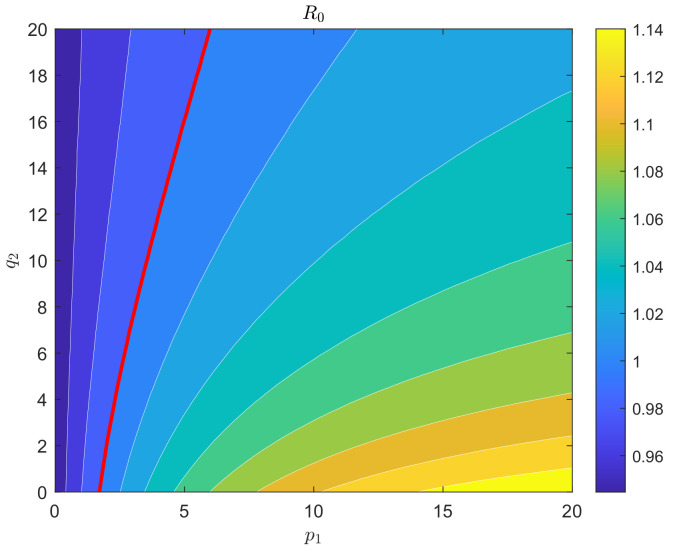
Contour plots of basic reproduction number $R_{0}$ with respect to preference parameters $p_{1}$ and $q_{2}$, showing that the values of $R_{0}$ increases as $p_{1}$ increases or as $q_{2}$ decreases.

**Figure 5 plants-12-03659-f005:**
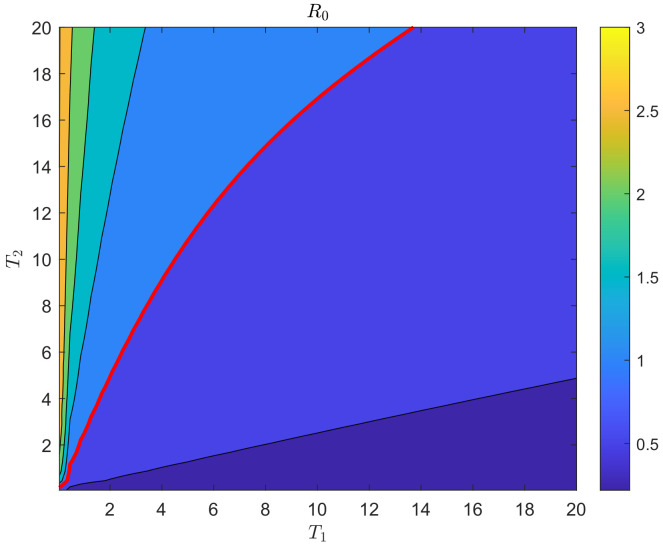
Contour plots of basic reproduction number $R_{0}$ vs. $T_{1}$ and $T_{2}$ showing the paired effects of the periods of effectiveness and ineffectiveness of the pesticide (the red line shows the threshold value where $R_{0}=1$). Different colors indicate increasing values of $R_{0}$ from 0 to 3 in increments of 0.5 (ranging from blue to yellow).

**Figure 6 plants-12-03659-f006:**
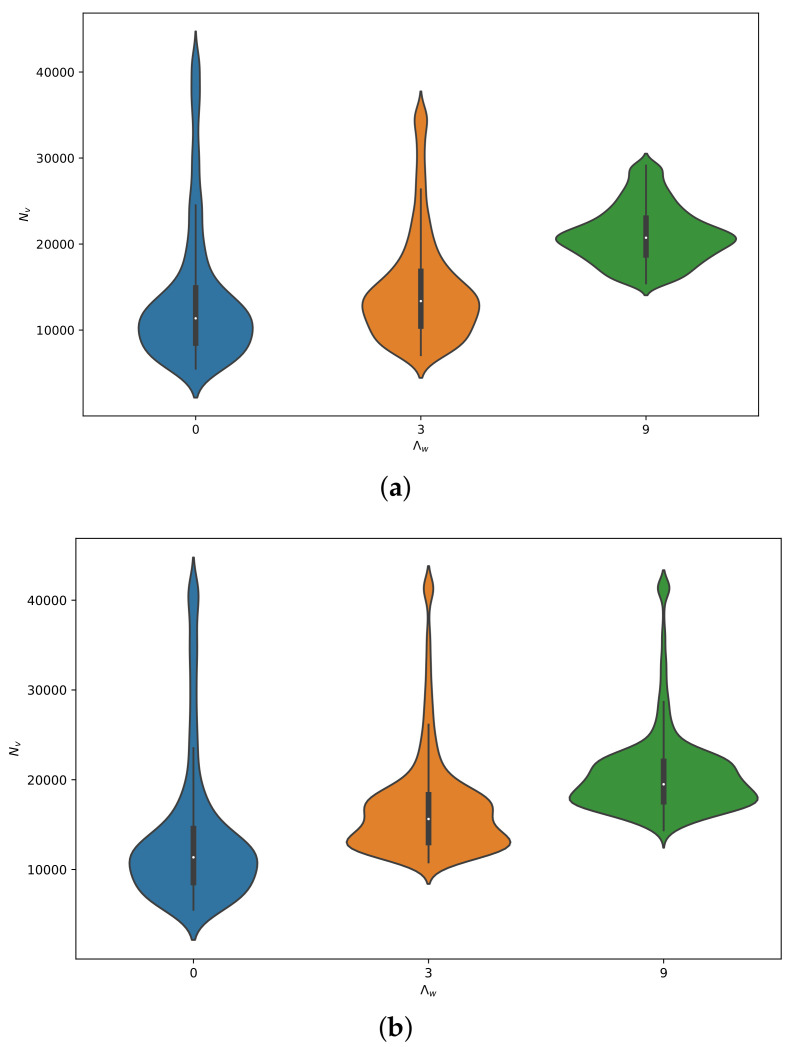
Violin Plots of total number of psyllids $N_{v}$ for three different constant recruitment rates $\Lambda_{w}$ with different infection rates, (**a**) $\beta_{1}=0.00494,\;\beta_{2}=0.00226$, (**b**) $\beta_{1}=0.01482,\;\beta_{2}=0.00678$. Showing that increasing the recruitment rate of the weed has a greater impact on the number of ACP.

**Table 1 plants-12-03659-t001:** Summary of the multi-host switching HLB model and its parameters (Equation 1) and (Equation 2).

Variable	Description
$\beta_{1}$	Probability that a susceptible citrus tree becomes infected from contact with ACPs infected virus
$\beta_{2}$	Probability that a susceptible ACP becomes infected from contact with an infected citrus tree
$\Lambda_{w}$	Constant recruitment rate for *A. conyzoides*
$\Lambda_{v}$	Constant recruitment rate of ACPs
$\mu_{h}$	Natural mortality of citrus trees
$\mu_{w}$	Mortality rate of *A. conyzoides*
γ	Rouging rate of infected trees
$d_{c}$	Natural mortality of ACPs in citrus tree
$d_{w}$	Natural mortality of ACPs in weeds
δ	Diffusion rate of ACPs
$p_{1}$	Bias parameter of ACPs from tree to *A. conyzoides* in the duration of effectiveness
$q_{1}$	Bias parameter of ACPs from *A. conyzoides* to tree in the duration of non-effectiveness
$p_{2}$	Bias parameter of ACPs from tree to *A. conyzoides* in the duration of effectiveness
$q_{2}$	Bias parameter of ACPs from *A. conyzoides* to tree in the duration of non-effectiveness
α	Growth rate parameter of *A. conyzoides* population
$\alpha_{1}$	Saturation effect parameter
θ	Killing rate of pesticide

**Table 2 plants-12-03659-t002:** Parameter values for the multi-host switching HLB model (Equation 1) and (Equation 2).

Parameter	Baseline Values	Unit	Reference
*K*	2000	-	[25]
$\beta_{1}$	0.00494	day${}^{-1}$	[26]
$\beta_{2}$	0.00226	day${}^{-1}$	[26]
$\Lambda_{v}$	924	day${}^{-1}$	[21]
$\Lambda_{w}$	3	day${}^{-1}$	Assumed
$\mu_{h}$	0.00011	day${}^{-1}$	[27]
$\mu_{w}$	0.00274	day${}^{-1}$	[28]
γ	0.001	day${}^{-1}$	[29]
$d_{c}$	0.0222	day${}^{-1}$	[30]
$d_{w}$	0.0333	day${}^{-1}$	[10]
δ	0.02	day${}^{-1}$	Assumed
$p_{1}$	3	day${}^{-1}$	Assumed
$p_{2}$	0	day${}^{-1}$	Assumed
$q_{1}$	0	day${}^{-1}$	Assumed
$q_{2}$	10	day${}^{-1}$	Assumed
$\alpha_{1}$	0.00003	day${}^{-1}$	Assumed
α	0.0015	day${}^{-1}$	[31]
θ	0.1454	day${}^{-1}$	[10]

## Data Availability

This research does not involve real data. No data were used, no new data are created.

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
