# Peer review of "Modeling Study of the Effects of Ageratum conyzoides on the Transmission and Control of Citrus Huanglongbing"

_plants, 2023, doi:10.3390/plants12203659_

Round 1

Reviewer 1 Report

In the paper  a mathematical model for the study of the effects of HLB disease, its transmission and control is proposed. In this model are incorporated the effects of A.conyzoides, vector preferences for settling and periodical pesticide application. The model is theoretically studied and numerical simulations confirm the obtained analytical results.

The paper is interesting both from theoretical and practical points  of view and may be published with some additional comments of the authors on the aspects presented in the attached file.

Author Response

Thank you for your positive review.

Reviewer 2 Report

The article is devoted to a current topic related to modeling the spread of pests in the Ageratum conyzoides environment. The authors point out that the modeling results lead to paradoxical conclusions. However, such conclusions may be questioned because there is no experimental evidence. If conducting a full-scale experiment is not possible, it would be useful to make a comparison with other known models, for example, the Lotka–Volterra mathematical model, or building machine learning models (if there is a sufficient set of data for training). From a mathematical point of view, the calculations given in the article are correct. In general, the article is of interest and can be published after eliminating the comments: it is necessary to argue why the model is correct and can be trusted without conducting a full-scale experiment.

Author Response

We sincerely thank the reviewers for their reading and comment on our paper. We have revised accordingly based on our answers to the questions from the reviewers below.

Reviewer 3 Report

see the attached report 

Just make sure your manuscript is proofread by the English native speaker. 

Author Response

(The authors gave the same response as above.)
